# Molecular phylogeny and cryptic morphology: A combined approach to taxonomic novelties in *Polycarpaea* (Caryophyllaceae) from Vietnam

Truong Thanh Hoang[1,2], Le Ba Le[1], Ngan Thi Kim Le[1], Minh Thi Ai Nguyen[1]*, Anh Thi Lan Truong[1], Nhung Thi Tran[1], Vinh Thai Tran[3], Son Van Le[4], Kim Thi Duong[1], Khoa Viet Bach Hoang[1], Trieu Ngoc Le[1], Binh Van Nguyen[1]*, Tien Van Tran[1]*

1 Dalat University, Dalat, Lam Dong Province, Vietnam, 2 Vietnam Forest Science Institute of Central Highlands and South of Central Vietnam, Vietnam Academy of Forest Science, Dalat, Lam Dong Province, Vietnam, 3 Taynguyen Institute for Scientific Research, Vietnam Academy of Science and Technology, Dalat, Vietnam, 4 Binh Chau-Phuoc Buu Nature Reserve, Xuyen Moc, Ba Ria Vung Tau Province, Vietnam

* minhnta@dlu.edu.vn (MTAN); binhnv@dlu.edu.vn (BVN); tientv@dlu.edu.vn (TVT)

**Data Availability Statement:** All relevant data are within the paper and its Supporting information files.

## Abstract

Three new species of *Polycarpaea* from Vietnam, *Polycarpaea vanphongensis* V.T. Tran, H.T. Truong & N.V. Binh, *Polycarpaea chungana* V.T. Tran, H.T. Truong & N.V. Binh, *Polycarpaea phuquocensis* V.T. Tran, H.T. Truong & N.V. Binh are described and illustrated based on evidence of molecular sequence data from two markers (ITS1-5.8S-ITS2 and *rps*16) and combined morphological characteristics. *Polycarpaea vanphongensis* is closely related to *Polycarpaea gaudichaudi* Gagnep., *Polycarpaea arenaria* (Lour.) Gagnep., *Polycarpaea phuquocensis* V.T. Tran, H.T. Truong & N.V. Binh but differs by its stem glabrous, leaf ovate to elliptic, glabrous, ovary oblong ovoid, base obtuse, apex attenuate, capsule oblong void, 3.8 mm long. *Polycarpaea phuquocensis* V.T. Tran, H.T. Truong & N.V. Binh differs from the three species mentioned above by its stem being densely villous, leaf spathulate, ciliate, ovary ovoid, base acute, apex obtuse, capsule ovoid, 1.2 mm long. *Polycarpaea chungana* V.T. Tran, H.T. Truong & N.V. Binh is most similar to *Polycarpaea lignosa* Gagnep., but differs in having leaf oblong or linear, sparse ciliate, sepal and petal apex deeply concaved or slightly bifid, ovary ovoid, ovoid, 0.8–1.0 mm long. Furthermore, the achievements of analysis using molecular data on the systematic positions of 7 other species are results that have not been in previous molecular analyses.

## Introduction

The perennial xerophytic genus *Polycarpaea* Lamrck (1792:3) [1] (Caryophyllaceae Juss) comprises approximately 50 species in the world, and 6 species have been recorded in Vietnam [2, 3]. It's widely distributed in the tropics and subtropics of the Old World, and a few occur in the New World tropics [3]. The systematics of *Polycarpaea* is still mainly based on morphological

**Funding:** This research was funded by grants from the Vingroup Innovation Foundation (VINIF.2020. DA04). The funders supported money for all the study and publishing but had no role in study design, data collection, and analysis, the decision to publish, or the preparation of the manuscript.

**Competing interests:** The authors have declared that no competing interests exist.

characters used for recognizing taxa and placed in the tribe Polycarpeae along with 15 other genera with capsular fruits [4, 5]. It is widely known, and aspects of its morphological characters are often considered to be diagnostic for an herbaceous annual or herbaceous perennial in habit. Notable plants belonging to *Polycarpaea* have very similar vegetative and reproductive characteristics, such as dichotomously branched herbs, sometimes woody at the base, leaves opposite and often with developed axillary buds, which give the presence of whorls, stipules scarious, inflorescences a cyme, terminal provided with scarious bracts, sepals scarious, petals structurally similar to the sepals, stamens 5, style simple, stigma punctate to globular, capsule 3-valved [6–12]. However, the characters used for the most common classification of *Polycarpaea* are based on characteristics of the stipules, petals, sepals, ovaries, and fruits.

During collecting trips along coastal and island foredunes in Vietnam from 2020–2022, the authors found several perennial populations of xerophytic Caryophyllaceae distributed on dunes along the coast. Specimens of branches and flowering branches were collected. Specimens of branches and flowering branches were collected and represented by the collection VinIF0312, VinIF0029, and VinIF0388 (Table 1). The ranges of the three collections fall into three regions. One of these is the population of collection VinIF0029 growing sparsely scattered in the grass vegetation on the backdune, which is the climatic type of semi-arid in the south center. The population of collection VinIF.0312 is growing sparsely scattered in the shrubby vegetation on the foredune, which is the climatic type of arid in the north-south. The population of collection VinIF.0388 is growing sparsely scattered in the casuarina plantation, which is the climatic type of semi-arid in Phu Quoc Island, South Vietnam. All collected specimens were dissected and studied. The analysis results show that vegetative and reproductive characteristics are similar to the abovementioned characteristics. However, there could not be keyed out to any named species from Vietnam and neighboring regions. Although it is often assumed that vegetative and reproductive characteristics should be considered when delimiting species. However, the interrelationships between species and their allies are too complex to resolve based on morphological characteristics alone. The evidence provided for achieving clear delimitation is the use of molecular data and morphological characters in investigating relationships within taxa. Recently, a phylogenetic reconstruction based on sequences of the nuclear ITS gene and chloroplast gene regions (*rps*16) has been extensively proven valuable for studies of Caryophyllaceae at low or high taxonomic levels [13–16]. Therefore, in this study, details of morphological characteristics were used for comparison and combined with phylogenetic analysis based on the nuclear ITS gene sequence and the chloroplast gene region

**Table 1. Taxon sampling information and DNA sequences used for phylogenetic analysis.**

| Species | Voucher | Locality | rps16 | ITS | References |
|---|---|---|---|---|---|
| *Polycarpaea stylosa* Gagn. | VinIF.0820 | Phu Hai District, Ha Tinh Province, Vietnam; $18^0201'454''$N, $106^0262'636''$E | PP475302 | PP472122 | This study |
| *Polycarpaea arenaria* Gagn. | VinIF.0777 | Trieu Phong District, Quang Tri Province, Vietnam; $16^0849'243''$N, $107^0252'699''$E | PP475303 | PP472117 | This study |
| *Polycarpaea gaudichaudii* Gagnep. | VinIF.1226 | Phu Loc District, Thua Thien Hue Province, Vietnam; $16^0271'798''$N, $108^0060'160''$E | PP475304 | PP472116 | This study |
| *Polycarpaea vanphongensis* V.T. Tran, H.T. Truong, N.V. Binh | VinIF.0029 | Ninh Hoa District, Khanh Hoa Province, Vietnam; $12^0637'956''$N, $109^0410'847''$E | PP475305 | PP472119 | This study |
| *Polycarpaea chungana* V.T. Tran, H.T. Truong, N.V. Binh | VinIF.0312 | Bac Binh District, Binh Thuan Province, Vietnam; $11^0126'395''$N; $108^0499'552''$E | PP475306 | PP472120 | This study |
| *Polycarpaea lignosa* Gagnep. | VinIF.1224 | Xuyen Moc District, Ba Ria Vung Tau Province, Vietnam; $10^030'09''$N, $107^030'46''$E | PP475307 | PP472121 | This study |
| *Polycarpaea phuquocensis* V.T. Tran, H.T. Truong, N.V. Binh | VinIF.0388 | Phu Quoc Island, Kien Giang Province, Vietnam; $10^0271'116''$N; $103^0926'947''$E | PP475308 | PP472118 | This study |

(*rps*16) to clarify the identities certainty of currently collected specimens of the genus *Polycarpaea* in Vietnam.

## Materials and methods

### Ethics statement

Plants studied were collected along coastal and island foredunes in Phu Hai District, Ha Tinh Province; Trieu Phong District, Quang Tri Province; Phu Loc District, Thua Thien Hue Province; Ninh Hoa District, Khanh Hoa Province; Bac Binh District, Binh Thuan Province; Xuyen Moc District, Baria—Vung Tau Province; Phu Quoc Island, Kien Giang Province. All locations are not protected in any way. The voucher specimens are deposited in the Dalat University Herbarium (DLU).

### Morphological observation

Vegetative parts and flowers were observed under an Olympus SZX2-ILLK light microscope and color photographs were taken with a Canon Power Shot SX10IS. Based on field investigation and specimen examination, all vegetative and reproductive characters of the plant material were checked taxonomic key and descriptions of *Polycarpaea*, which were provided from Vietnam [2, 6, 7]; from China [3]; from Malaysia [8]. The features of the new species were verified by consulting herbarium specimens at Herbaria (HN, VMN, IBSC, K, KUN, and P). Botanical terminology of the description was followed by Gagnepain [6, 7], Cowie [8], Ho [2], Bakker [9], and Arya [11, 12].

### Molecular markers

**Taxon sampling and DNA extraction.** The fresh leaves of the new species and their allies were collected and summarized in Table 1. They were used for DNA extraction using a modified cetyltrimethylammonium bromide (CTAB) method [17]. The quality and quantity of extracted genomic DNA were measured using a UV-spectrophotometer, and agarose-gel electrophoresis.

**PCR, PCR product purification, and sequencing.** The polymerase chain reaction (PCR) was performed in a 50 μl reaction mixture containing 25 μL My Red HS Taq mix (Bioline), 2.5 μL 10 pmol/μL forward primer, 2.5 μL 10 pmol/μL reverse primer, and approximately 50 ng of DNA templates. The PCR reaction was performed in thermocyclers (Eppendorf, Germany) with the following cycling parameters: 94˚C (5 min); 35 cycles of 94˚C (30 s), 55–60˚C (30 s); 72˚C (30 s), then 72˚C (7 min). The DNA sequence of the ITS1-5.8S rRNA-ITS2 region was isolated and amplified using the primer pairs ITS 5P (5′‒ `GGA AGG AGA AGT CGT AAC AAG G` ‒3′) and ITS 8P (5′‒ `CAC GCT TCT CCA GAC TAC A` ‒3′), respectively (Moller *et al.* 1997) [18]. Forward and reverse primers for *rps*16 intron isolation and amplification were *rps*F (5′‒ `GTG GTA GAA AGC AAC GTG CGA CTT`‒3′) and *rps*R2 (5′‒ `TCG GGA TCG AAC ATC AAT TGC AAC`‒3′), respectively [19]. PCR products were first visualized on agarose gels (1.0%) containing safe gel stain (Inclone, Korea), and then were purified using Isolate II PCR and Gel purification Kits (Bioline). DNA sequencing was performed using Sanger method using ABI 3730 sequencers (Singapore).

**Sequence alignment and phylogenetic analysis.** All sequences were assembled using SeqMan software (DNASTAR Lasergene, DNASTAR, USA) and then aligned using the MAFFT program [20]. Phylogenetic tree construction and the reliability assessment of internal branches were conducted using the maximum likelihood (ML) method with 1,000 bootstrap replicates using MEGA 6.0 [21].

**Nomenclature.** The electronic version of this article in Portable Document Format (PDF) in a work with ISSN or ISBN will represent a published work according to the International Code of Nomenclature of algae, fungi, and plants, and hence the new names contained in the electronic publication of a PLOS ONE article are effectively published under that Code from the electronic edition alone, so there is no longer any need to provide printed copies.

In addition, new names contained in this work have been submitted to IPNI, from where they will be made available to the Global Names Index. The IPNI LSIDs can be resolved and the associated information viewed through any standard web browser by appending the LSID contained in this publication to the prefix http://ipni.org/. The online version of this work is archived and available from the following digital repositories: PubMed Central, and LOCKSS.

## Results and discussions

### Phylogenetic relationships within *Polycarpaea* in Vietnam

The length of ITS1-5.8S-ITS2 regions in investigated taxa was 809–811 bp with 63 variable sites, while the length of the partial *rps* 16 gene was 882–885 with 8 variable sites. The combined sequence data sets (ITS1-5.8S-ITS2 and *rps*16) with 1693–1695 bp lengths and 71 variable sites were analyzed. The achieved phylogenetic relationship, dendrogram based on ML possessed the topologies with bootstraps values (MLBS)$\geq$ 65% mapped on the corresponding nodes (Fig 1).

In the strict phylogenetic relationship, dendrograms were consistently divided into two groups. The group with a long style can be recognized as having two subgroups. The first group, separated from the others with high bootstrap value (100%), included *Polycarpaea*

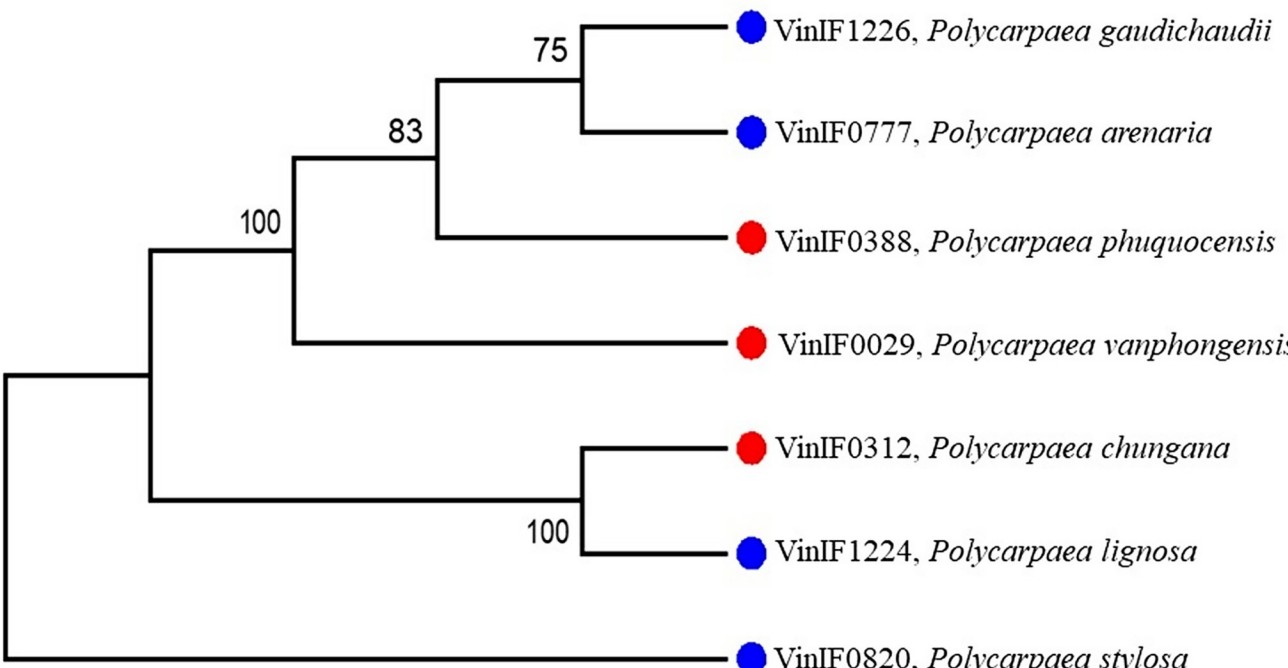

**Fig 1. The ML consensus tree of seven taxa based on two molecular markers (ITS1-5.8S-ITS2 and *rps*16).** Numbers at the nodes are ML bootstrap values from 1000 replicates. Red filled round bullets indicate new species in the current study, and blue filled round bullets indicate previously recorded species.

*gaudichaudii*, *P. areanria*, *P. vanphongensis*, and *P. phuquocensis*, and another comprising *P. lignosa* and *P. chungana* with high bootstrap support value (100%). The group with short style revealed that *P. stylosa* was a sister.

## Taxonomic implication of comparative molecular studies

As in previous analyses of the reproductive morphological characteristics of the genus *Polycarpaea* in Vietnam, Gagnepain [6, 7] indicated that species of this genus are divided into two groups; one group is characterized by a short style (*P. corymbosa* and *P. stylosa*) and the other group has a long style (*P. arenaria* and *P. gaudichaudii*). Most species that were currently collected from Vietnam were recovered within the group, which is the long style. Therefore, these new species are closely related to *P. arenaria*. *P. lignosa* and *P. gaudichaudii*. However, based on reproductive and vegetative characteristics, these new species are completely different.

*P. chungana* collected from Bac Binh, Binh Thuan Province is closely related to, but distinct from *P. lignosa* by its leaf oblong or linear, ciliate, sepal, and petal apex deeply concaved or slightly bifid. *Polycarpaea vanphongensis* collected from Van Ninh, Khanh Hoa Province, possesses a stem glabrous, leaf ovate to elliptic, glabrous, ovary oblong ovoid, base obtuse, apex attenuate. These characteristics can be distinguished considerably from *P. duongana*, *P. arenaria*, and *P. gaudichaudii*. *P. phuqocensis* collected from Phu Quoc island, Kien Giang Province, is similar to the three species mentioned above, but differs by leaf spathulate, ciliate, sepal and petal small, capsule ca. 1.2 mm long, stigma 1 mm long.

The molecular evidence (Fig 1), as well as morphological characteristics (Tables 2 and 3; Figs 2–8), indicated that *P. vanphongensis*, *P. chungana*, *P. phuquocensis* are closely related to, but differ from *P. arenaria*, *P. lignosa*, and *P. gaudichaudii*.

## Taxonomic treatment

***Polycarpaea vanphongensis*** V.T. Tran, H.T. Truong, N.V. Binh sp. nov.
[urn:lsid:ipni.org:names: 77342739–1] (Figs 3 and 4).

**Table 2. Morphological comparison between *Polycarpaea vanphongensis* sp. nov., *Polycarpaea phuquocensis* sp. nov. and *P. gaudichaudii*, *P. arenaria*.**

| *Characters* | *Polycarpaea vanphongensis* | *Polycarpaea phuquocensis* | *P. gaudichaudii* | *P. arenaria* |
|---|---|---|---|---|
| **Stem** | glabrous | densely villous | sparsely villous | densely villous |
| **Leaves** | ovate to elliptic, glabrous | spathulate, ciliate | narrowly oblong to oblong-elliptic, glabrous | linear, glabrous |
| **Stipules** | triangular, 1–1.5 mm long | triangular, 0.8–1.0 mm long, | triangular, 3–4 mm long | oblong acuminate, 3–4 mm long |
| **Inflorescence** | compound cyme, sparsely | compound cyme, sparsely | densely cyme | reniform densely |
| **Bracts** | ovate-lanceolate, 1.8–2.0 mm long, fimbriate along the margins, apex deeply bifid | triangular, 1.5–1.6 mm long, fimbriate along the margins, apex bifid or deeply bifid, | triangular, 3–4 mm long glabrous, apex acuminate | oblong, 3–4 mm long, apex acuminate |
| **Pedicel** | 2–3 mm long, glabrous | 5–13 mm long, densely villous | ca. 5 mm long, glabrous or ciliate | 2–3 mm long, dense ciliate |
| **Sepals** | oval, 2.8–3 mm long, ca. 1 mm wide, apex obtuse | oval ca. 2.2 mm long, ca. 1 mm wide, apex obtuse | oval lanceolate, ca. 3,5 mm long, 1.5 mm wide, apex obtuse | triangular, 3–3.5 mm long, 1 mm wide, apex obtuse |
| **Petals** | lanceolate 3.8–4 mm long, 1–1.2 mm wide | oval 2.9–3.1 mm long, 1.0–1.2 mm wide, apex obtuse | ovate-lanceolate, 4–5 mm long 1.5–2 mm wide, apex obtuse | triangular, 3–4.5 mm long, 1 mm wide, apex obtuse |
| **Ovary** | oblong ovoid, base obtuse, apex attenuate | ovoid, base acute, apex obtuse | ovoid, base acute, apex obtuse | ovoid, acute at base, obtuse at apex |
| **Style** | 2 mm long | 1 mm long | 0.5 mm long | 1.0–1.3 mm long |
| **Capsule** | oblong void, 3.8 mm long | ovoid, 1.2 mm long | ovoid, 3 mm long | ovoid, 3 mm long |

**Table 3. Morphological comparison between *Polycarpaea chungana* sp. nov. and *P. lignosa*.**

| Characters | *Polycarpaea chungana* | *P. lignosa* |
|---|---|---|
| **Leaves** | oblong or linear | ovoid or ellipsoid, |
| | ca. 1 cm long ca. 0.1 cm wide | ca. 0.8 cm long ca. 0.2 cm wide |
| | sparse ciliate | glabrous |
| | apex obtuse | apex acuminate |
| **Stipules** | ca. 3 mm long, ca.1.2 mm wide | ca. 2 mm long, ca. 1.5 mm |
| **Inflorescence** | | |
| **Bracts** | 2.5 mm long, 0.8 mm wide | 2.5 mm long, 0.8 mm wide |
| **Sepals** | ca. 2.2 mm long, ca. 1 mm wide, apex concave deeply or slightly bifid | 4 mm long, 1 mm wide apex obtuse |
| **Petals** | ca. 2.5 mm long, ca. 1 mm wide, apex deeply concaved or slightly bifid | 4 mm long, 0.8 mm wide, apex obtuse |
| **Ovary** | ovoid, 0.8–1.0 mm long | globose, 1–2 mm long |
| **Capsule** | ovoid | circular |

**Type.** VIETNAM, Khanh Hoa Province, Van Ninh district, Van Tho village, Van Phong Bay, Xuan Dung beach, coastal sand beach, alt. 5m, $109^0 40^{'} 85^{''}$E, $12^0 68^{'} 76^{''}$N, 9 September 2023, V. T. Tran, H. T. Truong, N. V. Binh VinIF.0029 (holotype: DLU), the same locally, 11 September 2023 V. T. Tran, H. T. Truong, N. V. Binh VinIF.1223 (isotypes: DLU).

**Diagnosis.** *Polycarpaea vanphongensis* is characterized by having a stem glabrous, leaf ovate to elliptic, glabrous, ovary oblong ovoid, base obtuse, apex attenuate, capsule oblong void, 3.8 mm long.

**Description.** Perennial herbs, sub-erect and branched at base, ca. 25 cm high. Stems terete, glabrous, nodes reddish swollen, internodes 2–4 cm long. Leaves at the base, pseudoverticillate, sessile, oval, plump, slightly revolute, glabrous, light green, 8–10 mm long, 4–5 mm wide, base rounded, margin recurved, apex apiculate, veins 2-paired, prominent veins on abaxial surface; leaves at the top ovate to elliptic, 4–7 mm long, 2–4 mm wide; stipules scarious, triangle shape, concave, 1–1.5 mm long, 1–1.5 mm wide, margin fimbriate, inner surface villous, outer surface prominently nerved, white. Inflorescence terminal or axils of terminal leaves, compound cyme sparsely ca. 2.5–5 cm long; bracts paired, ovate-lanceolate, fimbriate along the margin, apex deeply bifid, white, ca. 1.8–2.0 mm long. Flowers 4–4.5 mm long; bracteoles, triangle, fimbriate along the margin, bifid when old; pedicels 2–3 mm long, glabrous. Sepals 5, free, ovate 2.8–3 mm long, ca. 1 mm wide, margin entire, apex acute, a yellowish circle around a reddish pot at the base, dark red when old, midrib prominent. Petals 5, lanceolate 3.8–4 mm long, 1–1.2 mm wide, margin entire, acute at apex, embraced the ovary, 1/3 longer than sepals, dark red at the base and along the midrib to 3/4 petals. Stamens 5, villous filament at the base, ca. 3–3.5 mm long; anthers oblong. Ovary 1-loculed, long ovoid, apex obtuse, apex attenuate 0.8–1.0 mm long, 0.4–0.5 mm wide; style 2 mm, slender; stigma lobbed. Capsule ovoid, 3.6–3.8 mm long, 1.2–1.5 mm wide, shortly stipitate, 3-valved, breaks along the suture, brownish, thin along the margin.

**Etymology.** The specific epithet refers to the type locality, Van Phong Bay, Van Ninh district, Khanh Hoa Province.

**Phenology.** The plants were flowering and fruiting between September and November.

**Additional specimens examined.** Vietnam. *Polycarpaea arenaria* (*Polycapaea arenaria* var. *condorensis*): Indochine, Iles de Poulo-Condor (Cochinchine française), 1876–8, F.J. Harmand 662 (P01902862!, P01902863!); *Polycarpaea arenaria* (*Polycarpaea arenaria* var. *longifolia*). Cambodia monts de Knang-Krepeuh, May 1870, Pierre (P04925647!, P04925648!);

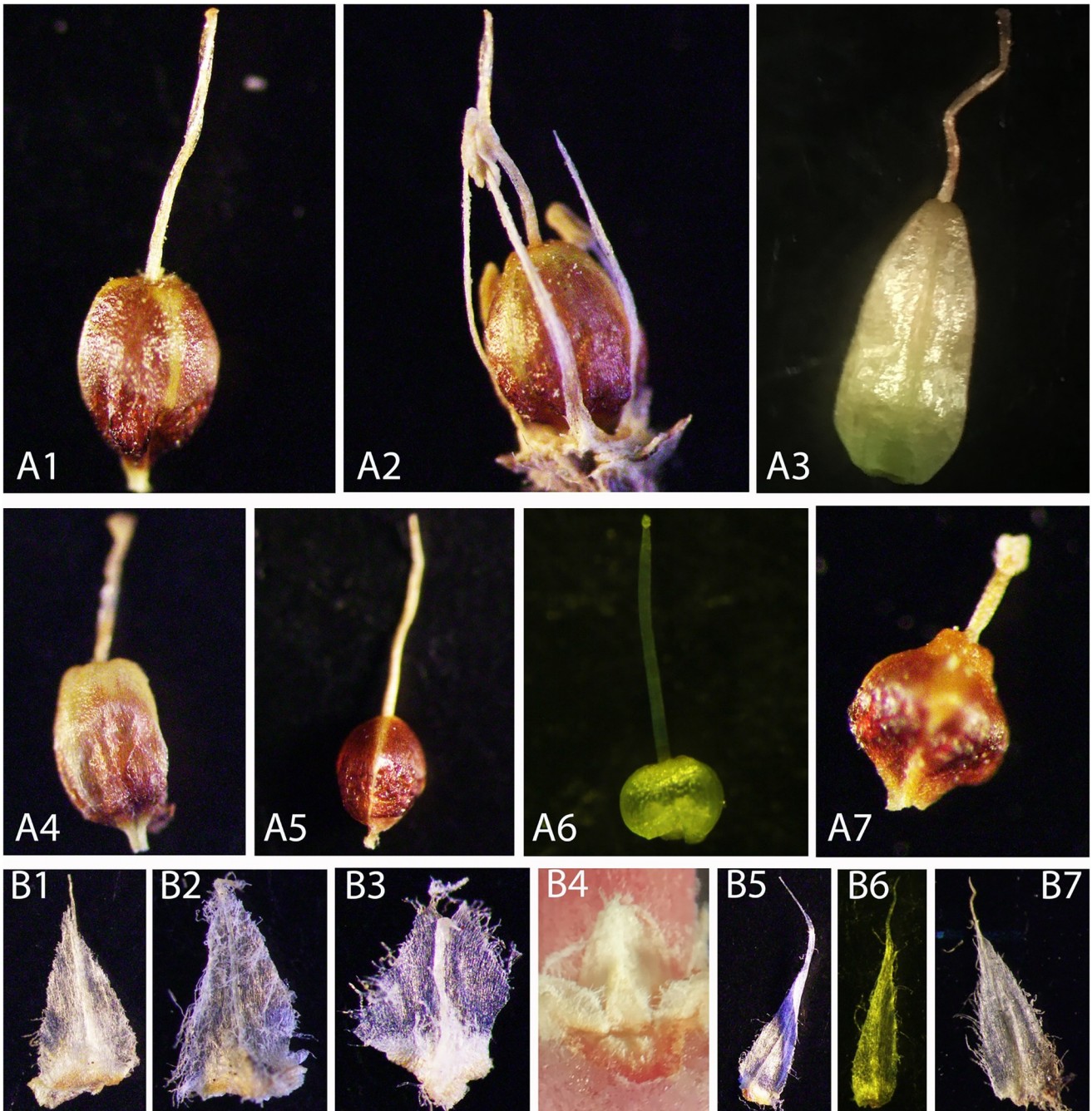

**Fig 2. Comparison of reproductive characters.** Ovaries: A1. *P. gaudichaudii*; A2. *P. arenaria*. A3. *P. vanphongensis*; A4. *P. phuquocensis*; A5. *P. chungana*; A6. *P. lignosa*; A7. *P. stylosa*. Stipules: B1. *P. gaudichaudii*; B2. *P. arenaria*; B3. *P. phuquocensis*; B4. *P. vanphongensis*; B5. *P. chungana*; B6. *P. lignosa*; B7. *P. stylosa*. Photographs by V.T. Tran, T.T. Vinh.

Indochine, Pro. Ba Ria, Xuyen-mot, Feb. 1866, sept. 1860, sept. 1864, Mar 1868, Pierre (P04925632!, P04925633!, P04925636! P04925645!); Pro. Bien Hoa, Bao Chanh, sept. 1867, Pierre (P04925636!); *Polycarpaea arenaria* Gagnep. (*Polycarpaea arenaria* var. *parviflora*). Vietnam. Indo-Chine, Cochinchine, La-Thien, Bao-Chiang, Sept. 1865 (1862-8/1866-8), C. Thorel 1966 (P01902866!, P01902867!, P01902868!,); *Polycarpaea arenaria* Annam: Ca Na,

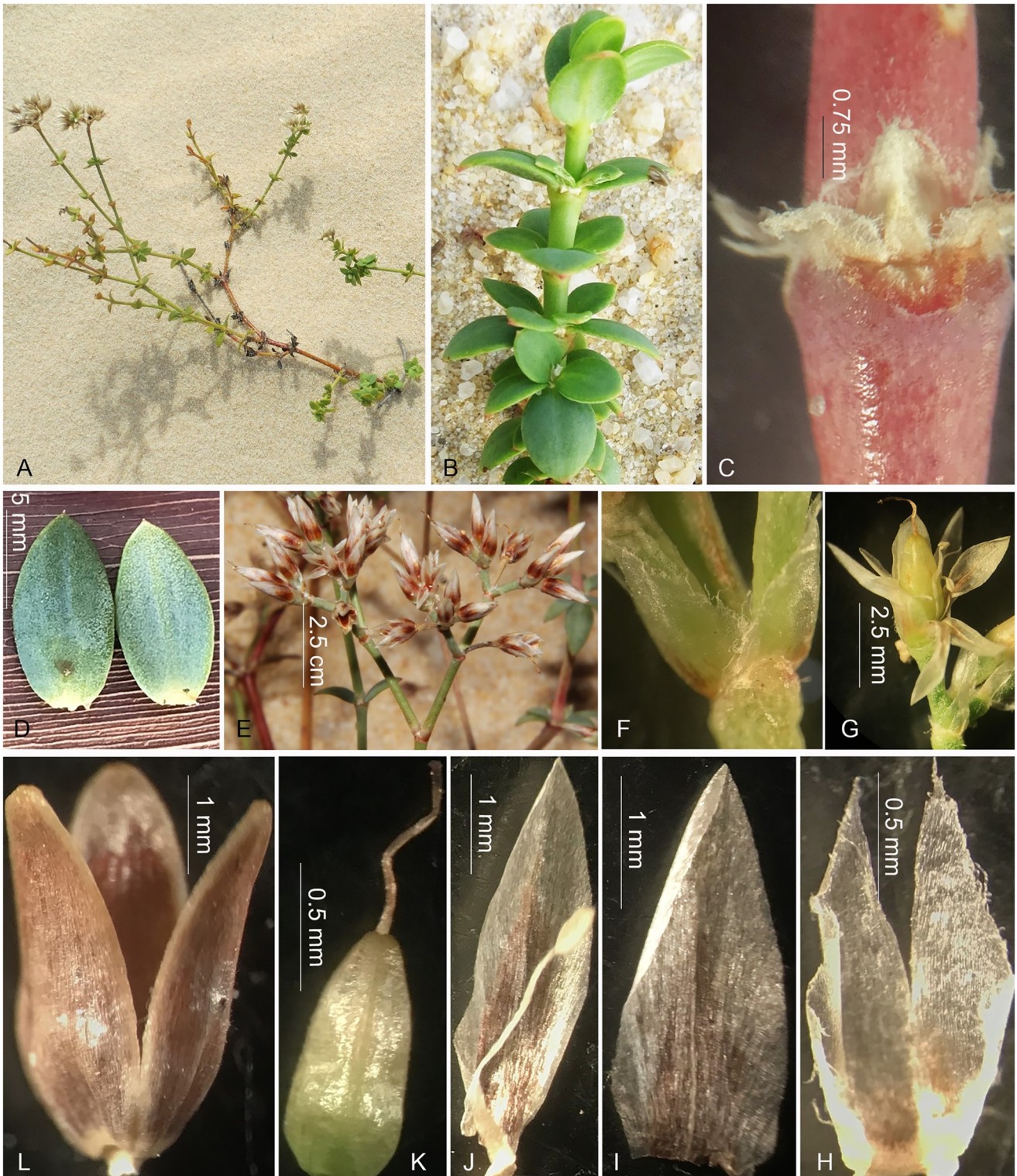

**Fig 3. Field photos of *Polycarpaea vanphongensis* V.T. Tran, H.T. Truong, N.V. Binh sp. nov.** A. Habit. B. Section of the stem with pseudoverticillate leaves. C. Internode with stipules. D. Leaves. E. Inflorescence. F, H. Bracts. G. Flower cluster. I. Sepal. J. Petal with stamen. K. Gynoecium. L. Capsule. Photo by V. T. Tran, H. T. Truong.

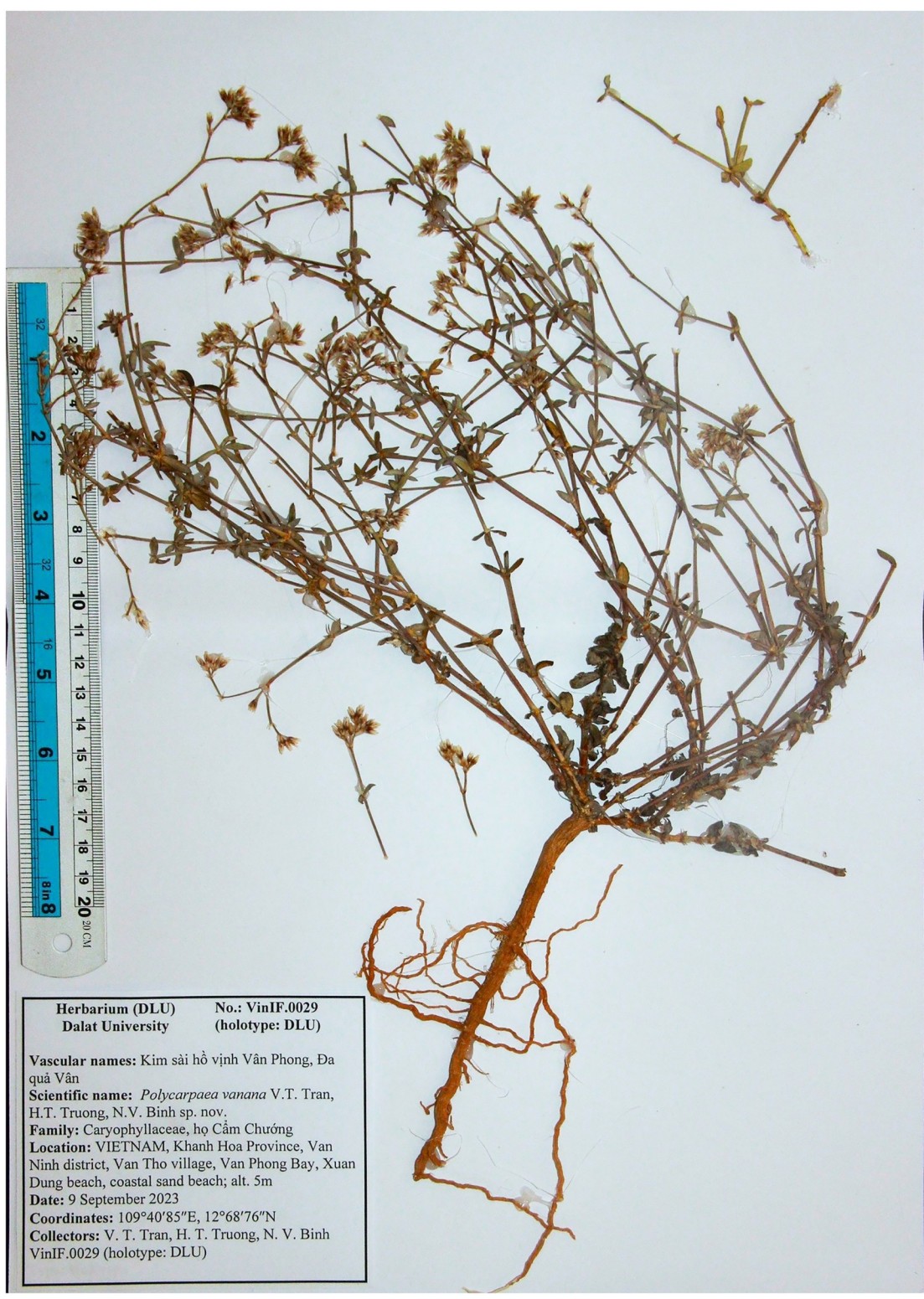

**Fig 4. Holotype of *Polycarpaea vanphongensis* V.T. Tran, H.T. Truong, N.V. Binh.**

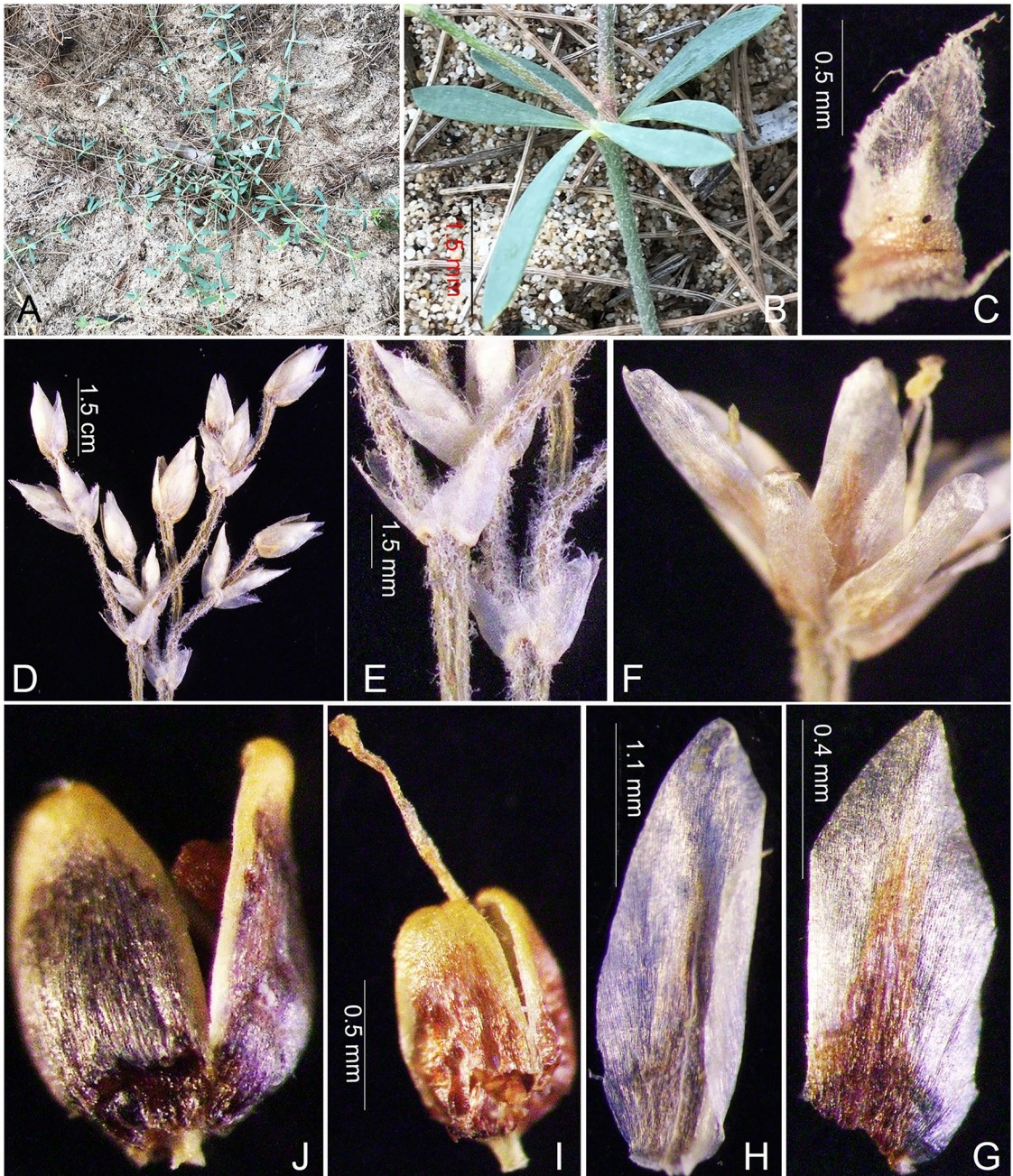

**Fig 5. Field photos of *Polycarpaea phuquocensis* V.T. Tran, H.T. Truong, N.V. Binh sp. nov.** A. Habit; B. Section of the stem with pseudoverticillate leaves; C. Stipule; D. Inflorescence. E. Bracts; F. Opening Flower; G Sepal; H. Petal; I. Dry Gynoecium; L. Capsule. Photo by V. T. Tran, T. T. Vinh.

Pro: Phanrang, 24-12-23, Poilane 9297 (P04925641!); *Polycarpaea arenaria* Ca Na, Pro: Phan-rang, 3-3-23, Poilane 5531 (P04925638!);

*Polycarpaea phuquocensis* V. T. Tran, H.T. Truong & N. V. Binh sp. nov.
[urn:lsid:ipni.org:names: 77342740–1] (Figs 5 and 6).

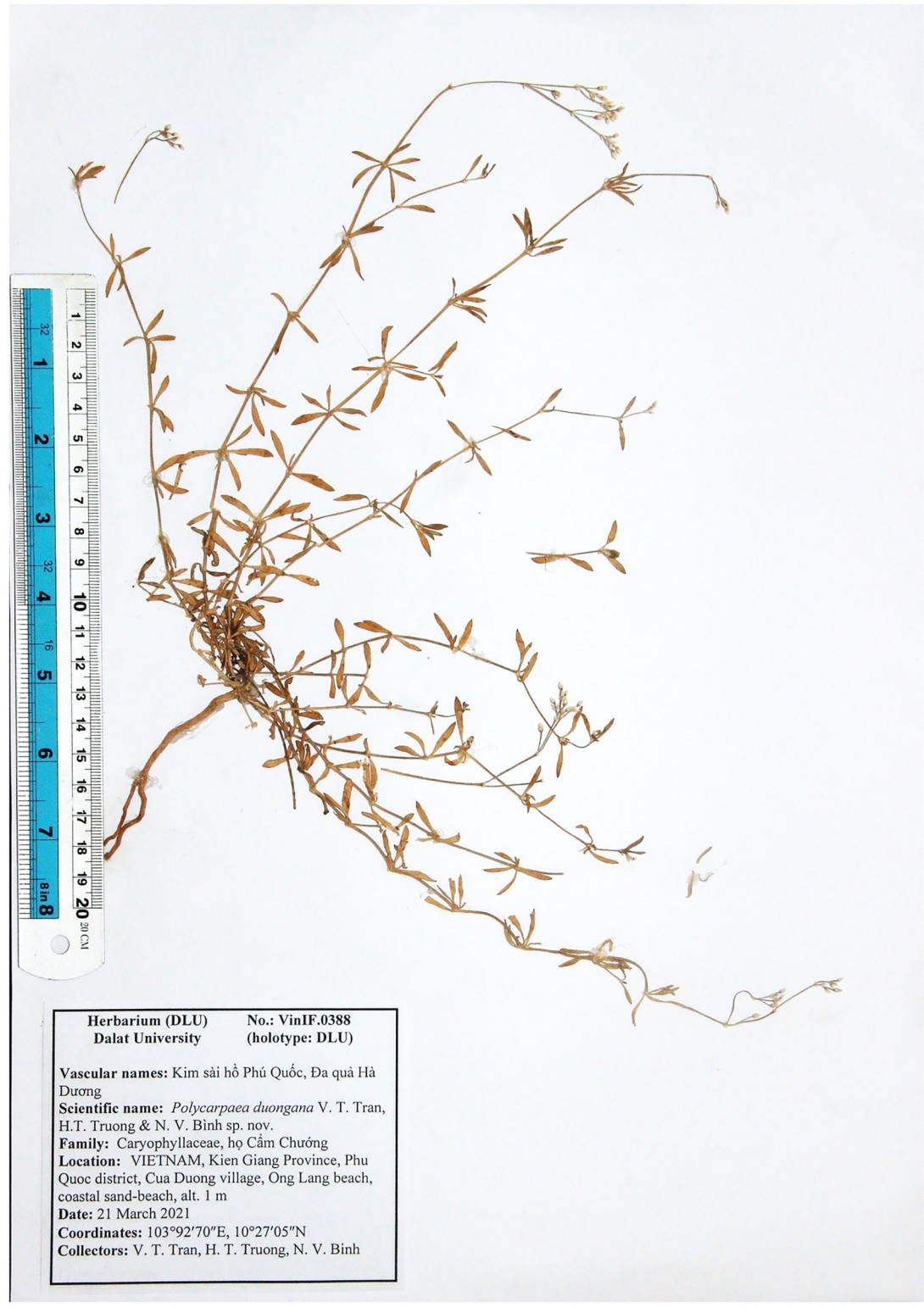

**Herbarium (DLU)** **No.: VinIF.0388**
**Dalat University** **(holotype: DLU)**

**Vascular names:** Kim sài hồ Phú Quốc, Đa quả Hà
Dương
**Scientific name:** *Polycarpaea duongana* V. T. Tran,
H.T. Truong & N. V. Bình sp. nov.
**Family:** Caryophyllaceae, họ Cẩm Chướng
**Location:** VIETNAM, Kien Giang Province, Phu
Quoc district, Cua Duong village, Ong Lang beach,
coastal sand-beach, alt. 1 m
**Date:** 21 March 2021
**Coordinates:** 103°92′70″E, 10°27′05″N
**Collectors:** V. T. Tran, H. T. Truong, N. V. Binh

**Fig 6. Holotype of *Polycarpaea phuquocensis* V.T. Tran, H.T. Truong, N.V. Binh.**

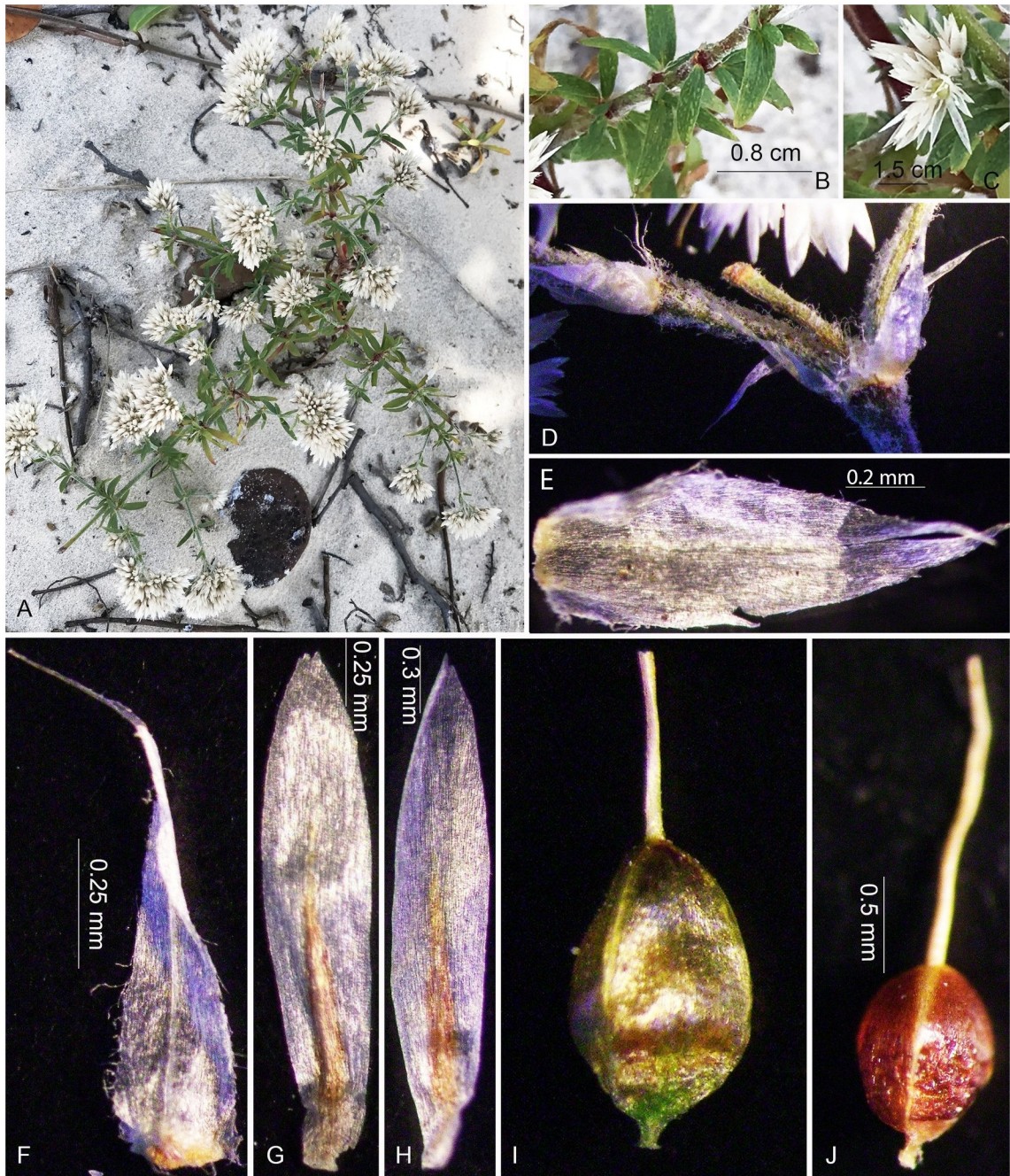

**Fig 7. Field photos of *Polycarpaea chungana* V.T. Tran, H.T. Truong, N.V. Binh sp. nov.** A. Habit; B. Section of the stem with pseudoverticillate leaves; C. Flowers; D-E. Bracts; F. Stipule; G. Sepal; H. Petal; I. Fresh gynoecium; J. Dry Gynoecium. Photo by V. T. Tran, T. T. Vinh.

**Type.**   VIETNAM, Kien Giang Province, Phu Quoc district, Cua Duong village, Ong Lang beach, coastal sand-beach, alt. 1 m, $103^0 92' 70''$E, $10^0 27' 05''$N, 21 March 2021, V. T. Tran, H. T. Truong, N. V. Binh VinIF.0388 (holotype: DLU).

**Diagnosis.**   *Polycarpaea phuquocensis* is characterized by having a stem densely villous, leaf spathulate, ciliate, ovary ovoid, base acute, apex obtuse, capsule ovoid, 1.2 mm long.

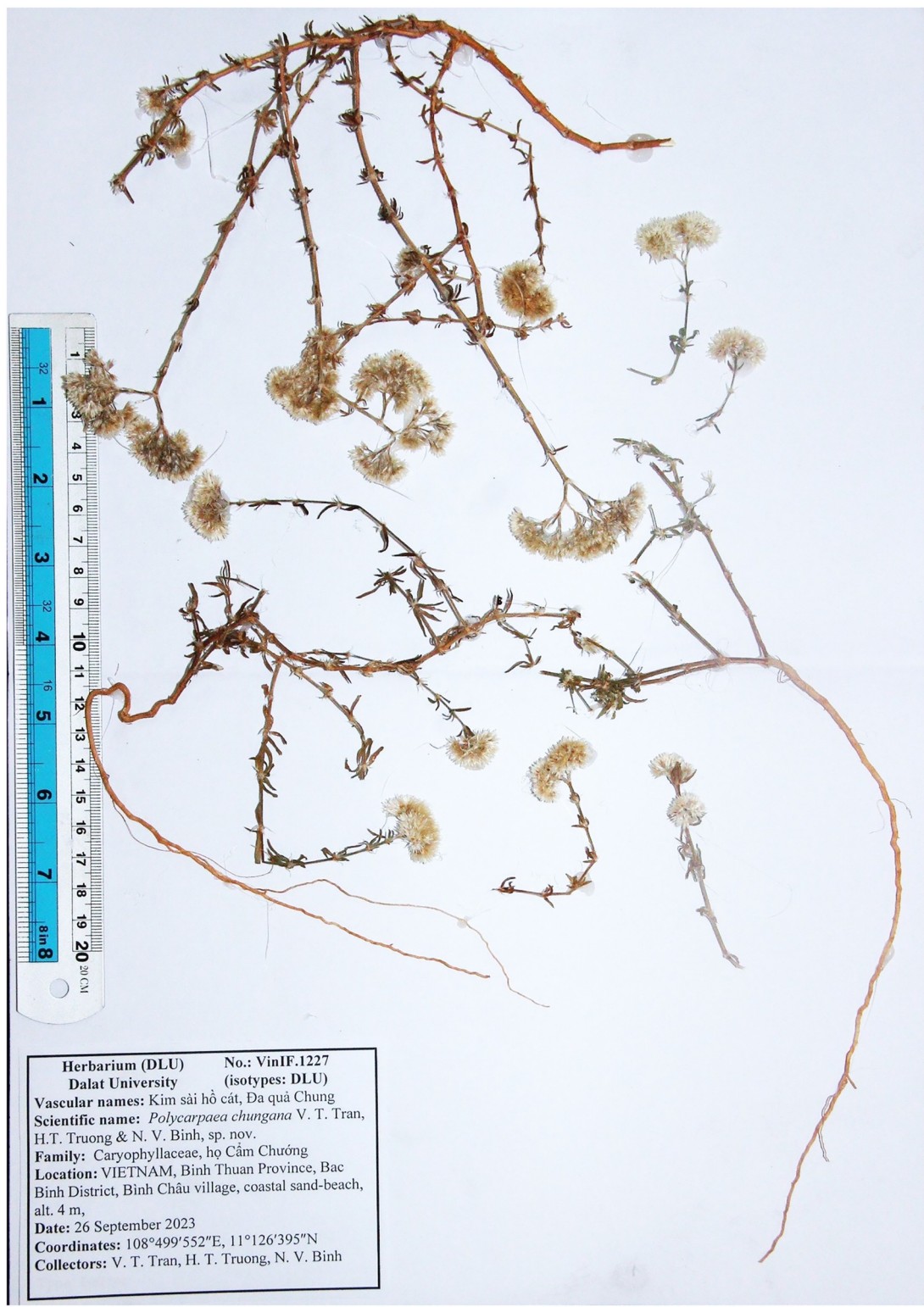

**Fig 8. Holotype of *Polycarpaea chungana* V.T. Tran, H.T. Truong, N.V. Binh.**

**Description.** Perennial herbs, sub-creeping and branched at base, 20–25 cm long. Stems terete, densely villous, nodes greenish, internodes 2–3 cm long. Leaves spathulate, pseudoverticillate to opposite, ciliate, 1.6–1.8 long, 3.0 mm wide; leaves at top: rare lanceolate, light green, 1.5–1.8 mm long, 1.9–2.1 mm wide, base truncate, entire margins, apex apiculate, 1 midrib and prominent on abaxial surface; stipules scarious, triangle, 0.8–1.0 mm long, 0.4–0.6 mm wide, apex apiculate ca. 0.3 mm, margins fimbriate, villous, outer surface prominently nerved, white. Inflorescence terminal, compound cyme sparsely with a 3.0–5.0 cm long; bracts paired, scarious, triangle, apex apiculate to attenuate, fimbriate along the margins, bifid or slightly bifid, white, ca. 1.5–1.6 mm long, 0.8–1.5 mm wide. Flowers 2.8–3.0 mm long, 0.8–2.0 mm white; pedicels 5–13 mm long, densely villous. Sepals 5, free, oval ca. 2.2 mm long, 1 mm wide, sparse fimbricate at the base, apex obtuse. Petals 5, oval 2.9–3.1 mm long, 1.0–1.2 mm wide, obtuse, margins entire, apex acute, embraced ovary, ca. 3/4 longer than sepals, lemon along the midrib from base to 3/4 of petals. Stamens 5, glabrous, ca. 2.2 mm long; anthers yellow, oblong, dorsifixed. Ovary 1-loculed, shortly stipitate, ovoid, 0.8–1.0 mm long, 0.7–1.0 mm wide, finely lined, placentation free central; style 1 mm. Capsule ovoid 1.0–1.2 mm long, 0.5–0.7 mm wide, 3-valved, breaks along the suture, brownish, thin along the margin.

**Etymology.** The specific epithet refers to the type locality, Phu Quoc island, Kiên Giang Province.

**Phenology.** The plants were flowering and fruiting between March and June.

**Additional specimens examined.** *Polycarpaea gaudichaudii.* Mallaca, Voyage de M. Gaudichaud sur la Bonite 1836–37, Feb. 1837, s.n. 184 (P01902871!, P01902872!); *Polycarpaea gaudichaudii.* Vietnam. Cochinchine-Tourane, Voyage de M. Gaudichaud sur la Bonite 1836–37, Jan. 1837, s.n. 184 (P01902869!, P01902870!); *Polycarpaea gaudichaudii.* Annam. Tourane, 17-2-39, Poilane 28844 (dupl. VNM, E), Poilane 28844 (P05139175!, P04925935!, P04925942!); *Polycarpaea gaudichaudii.* Cochinchine, Phú Quốc, Dương Đông, 13-6-1938, E. Poilane 27275 (P04925936!); *Polycarpaea gaudichaudii.* Indochine, Tourane, 1912, H. Lecomte and A. Finet 874 (P04925937!); *Polycarpaea gaudichaudii.* Cochinchine, Nha Trang, Bá-Ta, 23-4-23, Poilane 6045 (P04925939!); *Polycarpaea gaudichaudii.* Indochine, Tourane, 12-II-II, Lazaret 902 (P04925941!); *Polycarpaea gaudichaudii.* Indo-chine, Tourane, 19 Jan. 1903, D. Bois 695 (P04925943!); *Polycarpaea gaudichaudii.* Indo-chine, Thừa Thiên Huế, Plain de Nước-Ngọt, 28 Juin 1916, s.n. 3197 (P04925948!).

*Polycarpaea chungana* V. T. Tran, H.T. Truong & N. V. Binh, sp. nov.

[urn:lsid:ipni.org:names: 77342741–1] (Figs 7 and 8).

**Type.** VIETNAM, Binh Thuan Province, Bac Binh District, Bình Châu village, coastal sand-beach, alt. 4 m, $108^0499^{'}552^{''}$E, $11^0126^{'}395^{''}$N, August 6, 2023, V. T. Tran, H. T. Truong, N. V. Binh VinIF.0312 (holotype: DLU), the same locally, September 26, 2023 V. T. Tran, H. T. Truong, N. V. Binh VinIF.1227 (isotypes: DLU).

**Diagnosis.** *Polycarpaea chungana* is characterized by having leaf oblong or linear, sparse ciliate, sepal, and petal apex deeply concaved or slightly bifid, ovary ovoid, ovoid, 0.8–1.0 mm long.

**Description.** Perennial herbs, sub-creeping and branched at base, 25–35 cm long. Stems terete, nodes greenish, internodes 1.2–1.5 cm long, densely villous. Leaves pseudoverticillate, sessile, oblong or linear, 0.6–0.8 cm long, ca. 0.1 cm wide; margins entire, apex acute or obtuse, sparse white ciliate, 1 midrib, prominent midrib on abaxial surface; stipules scarious, oblong triangle, 2–3 mm long, 1–1.2 mm wide, apex apiculate ca. 3 mm, margin fimbriate, outer surface prominently nerved, white. Inflorescence terminal, compound cyme, densely 1.5–2.0 cm long; bracts paired, scarious, oblong triangle, 2–2.5 mm long, 0.8–1.0 mm wide, apex apiculate, ca. 2 mm, margin fimbriate, outer surface prominently nerved, white. Flowers 2.8–3.0 mm long, 0.8–2.0 mm white; pedicels 2–3 mm long, densely long villous. Sepals 5, free, oblong-

ovate ca. 2.2 mm long, 1 mm wide, sparse fimbricate at the base, apex concave deeply or slightly bifid. Petals 5, oblong-ovate 2.0–2.5 mm long, 1.0–1.2 mm wide, apex concave deeply or slightly bifid, margins entire, embraced ovary, orange along the midrib from base to 3/4 of petals. Stamens 5, villous, ca. 2.2 mm long; anthers yellow, oblong, dorsifixed. Ovary 1-loculed, shortly stipitate, ovoid, 0.8–1.0 mm long, 0.7–1.0 mm wide, finely lined, placentation free central; style 1 mm. Capsule ovoid, 1.0–1.2 mm long, 0.5–0.7 mm wide, 3-valved, breaks along the suture.

**Etymology.**   *Polycarpaea chungana* is named in honor of Mr. Tran Kim Chung for his kind of grant for the first author (Ph.D. Student) study in the genus *Polycarpeae* from Vietnam.

**Phenology.**   The plants were flowering and fruiting between August and November.

**Additional specimens examined.**   *Polycarpaea lignosa*. Vietnam. Indochine, Phan Thiết, route de Phu-hai, 5 Nov. 1924, F. Evrard, 1720 (P01902864!, P01902865!).

## Conclusions

Based on the molecular data with ITS1-5.8S-ITS2 region and *rps*16 gene and morphology, a combined approach to taxonomic novelties in *Polycarpaea* (Carophyllaceae) from Vietnam. Three new species, *P. vanphongensis* and *P. duongana*, *P. chungana*, are sparsely distributed along the coast and Island of South VietNam. The primary habitat of new *Polycarpaea* is the coastal dunes have an arid climate and are associated with *Hedyostis pinifolia*, *Fimbristylis lasiophylla*, *Glinus oppositifolius*, *Euphorbia atoto*, *Gisekia pharmacoides*, *Ipomaea stolonifera*, *Ipomoea pes-caprae*, *Launaea sarmentosa*. The result also suggests that integrated morphological and molecular data should used to recognize and describe new species of *Polycarpaea*.

## Supporting information

**S1 Fig. The ML consensus tree of two molecular marker rps16.**
(TIF)

**S2 Fig. The ML consensus tree of marker ITS1-5.8S-ITS2.**
(TIF)

**S3 Fig. The quality of total extracted genomic DNA and PCR products.** a. total extracted genomic DNA. b. ITS1-5.8S rRNA-ITS2 PCR products. c. rps16 intron PCR products. M. DNA ladder (Bioline a. 1kb. b. c. 100bp).
(TIF)

**S4 Fig. The quality of DNA sequencing products.** A, ITS1-5.8S rRNA-ITS2 sequences; B, rps16 intron sequences.
(TIF)

**S1 Table. The quality and quantity of total extracted genomic DNA.**
(DOCX)

## Author Contributions

**Conceptualization:** Truong Thanh Hoang, Le Ba Le, Ngan Thi Kim Le, Minh Thi Ai Nguyen, Anh Thi Lan Truong, Nhung Thi Tran, Vinh Thai Tran, Son Van Le, Kim Thi Duong, Khoa Viet Bach Hoang, Trieu Ngoc Le, Binh Van Nguyen, Tien Van Tran.

**Data curation:** Truong Thanh Hoang, Minh Thi Ai Nguyen, Anh Thi Lan Truong, Nhung Thi Tran, Kim Thi Duong, Trieu Ngoc Le, Binh Van Nguyen, Tien Van Tran.

**Investigation:** Truong Thanh Hoang, Le Ba Le, Minh Thi Ai Nguyen, Anh Thi Lan Truong, Nhung Thi Tran, Vinh Thai Tran, Son Van Le, Kim Thi Duong, Khoa Viet Bach Hoang, Binh Van Nguyen, Tien Van Tran.

**Methodology:** Truong Thanh Hoang, Le Ba Le, Minh Thi Ai Nguyen, Binh Van Nguyen, Tien Van Tran.

**Software:** Minh Thi Ai Nguyen, Binh Van Nguyen, Tien Van Tran.

**Visualization:** Vinh Thai Tran, Khoa Viet Bach Hoang, Binh Van Nguyen, Tien Van Tran.

**Writing – original draft:** Truong Thanh Hoang, Minh Thi Ai Nguyen, Binh Van Nguyen.

**Writing – review & editing:** Tien Van Tran.

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
