## [Decision Letter · Decision Letter 0]

18 Apr 2024

PONE-D-24-10601Molecular Phylogeny and Cryptic Morphology: A Combined Approach to Taxonomic Novelties in Polycarpaea (Caryophyllaceae) from VietnamPLOS ONE

Dear Dr. Tran,

Thank you for submitting your manuscript to PLOS ONE. After careful consideration, we feel that it has merit but does not fully meet PLOS ONE’s publication criteria as it currently stands. Therefore, we invite you to submit a revised version of the manuscript that addresses the points raised during the review process. Please submit your revised manuscript by Jun 02 2024 11:59PM. If you will need more time than this to complete your revisions, please reply to this message or contact the journal office at plosone@plos.org. Please include the following items when submitting your revised manuscript:A rebuttal letter that responds to each point raised by the academic editor and reviewer(s). You should upload this letter as a separate file labeled 'Response to Reviewers'.A marked-up copy of your manuscript that highlights changes made to the original version. You should upload this as a separate file labeled 'Revised Manuscript with Track Changes'.An unmarked version of your revised paper without tracked changes. You should upload this as a separate file labeled 'Manuscript'.If applicable, we recommend that you deposit your laboratory protocols in protocols.io to enhance the reproducibility of your results. Protocols.io assigns your protocol its own identifier (DOI) so that it can be cited independently in the future. For instructions see: https://journals.plos.org/plosone/s/submission-guidelines#loc-laboratory-protocols. Additionally, PLOS ONE offers an option for publishing peer-reviewed Lab Protocol articles, which describe protocols hosted on protocols.io. Read more information on sharing protocols at https://plos.org/protocols?utm_medium=editorial-email&utm_source=authorletters&utm_campaign=protocols.

We look forward to receiving your revised manuscript.

Kind regards,

Waqas Khan Kayani, PhD

Academic Editor

PLOS ONE

Journal Requirements:

"This research was funded by grants from the Vingroup Innovation Foundation (VINIF.2020.DA04)."

"This research was funded by grants from the Vingroup Innovation Foundation (VINIF.2020.DA04). The first author (Ph.D. Student) would like to thank Mr. Tran Kim Chung and CT Group for their kindness in granting scholarships."

Please note that funding information should not appear in the Acknowledgments section or other areas of your manuscript. We will only publish funding information present in the Funding Statement section of the online submission form. Please remove any funding-related text from the manuscript. 

5. Thank you for stating the following in your Competing Interests section: "No"

6. Please provide a complete Data Availability Statement in the submission form, ensuring you include all necessary access information or a reason for why you are unable to make your data freely accessible. If your research concerns only data provided within your submission, please write "All data are in the manuscript and/or supporting information files" as your Data Availability Statement.

7. When completing the data availability statement of the submission form, you indicated that you will make your data available on acceptance. We strongly recommend all authors decide on a data sharing plan before acceptance, as the process can be lengthy and hold up publication timelines. Please note that, though access restrictions are acceptable now, your entire data will need to be made freely accessible if your manuscript is accepted for publication. This policy applies to all data except where public deposition would breach compliance with the protocol approved by your research ethics board. If you are unable to adhere to our open data policy, please kindly revise your statement to explain your reasoning and we will seek the editor's input on an exemption. Please be assured that, once you have provided your new statement, the assessment of your exemption will not hold up the peer review process.

8. We note that Figure 1 in your submission contain map images which may be copyrighted. All PLOS content is published under the Creative Commons Attribution License (CC BY 4.0), which means that the manuscript, images, and Supporting Information files will be freely available online, and any third party is permitted to access, download, copy, distribute, and use these materials in any way, even commercially, with proper attribution. For these reasons, we cannot publish previously copyrighted maps or satellite images created using proprietary data, such as Google software (Google Maps, Street View, and Earth). For more information, see our copyright guidelines: http://journals.plos.org/plosone/s/licenses-and-copyright.

1) You may seek permission from the original copyright holder of Figure 1 to publish the content specifically under the CC BY 4.0 license.  

2) If you are unable to obtain permission from the original copyright holder to publish these figures under the CC BY 4.0 license or if the copyright holder’s requirements are incompatible with the CC BY 4.0 license, please either i) remove the figure or ii) supply a replacement figure that complies with the CC BY 4.0 license. Please check copyright information on all replacement figures and update the figure caption with source information. If applicable, please specify in the figure caption text when a figure is similar but not identical to the original image and is therefore for illustrative purposes only.

9. We are unable to open your Supporting Information file [Supporting Information.rar]. Please kindly revise as necessary and re-upload.

Reviewers' comments:

Reviewer's Responses to Questions

**Comments to the Author**

1. Is the manuscript technically sound, and do the data support the conclusions?

Reviewer #1: Partly

Reviewer #2: Yes

2. Has the statistical analysis been performed appropriately and rigorously? 

Reviewer #1: N/A

Reviewer #2: N/A

3. Have the authors made all data underlying the findings in their manuscript fully available?

Reviewer #1: No

Reviewer #2: Yes

4. Is the manuscript presented in an intelligible fashion and written in standard English?

Reviewer #1: No

Reviewer #2: Yes

5. Review Comments to the Author

Reviewer #1: Manuscript: Molecular Phylogeny and Cryptic Morphology: A Combined Approach to Taxonomic Novelties in Polycarpaea (Caryophyllaceae) from Vietnam

The manuscript is aimed at describing the Molecular Phylogeny and Cryptic Morphology of genus Polycarpaea (Caryophyllaceae) from Vietnam.

Critical aspects of the manuscript are:

• In Abstract the names of new species should be written in full form along with their author citations.

• Keywords are missing from manuscript.

• Introduction is very poor (same consideration for the Abstract), it should be enriched consulting all the floristic works, at national or regional scale, covering the area as well as the taxonomic papers on Caryophyllaceae. There is no need to write the collection methodology in introduction section. Purpose of study is not mentioned along with poorly written objectives of research. Introduction also lacks latest references.

• In methodology section the ecology of species collected is missing. There is no information mentioned regarding phenology of the species season of collection and condition of soil during plant collection. In line number 73 it is mentioned that morphological characters are studied under light microscope. According to my knowledge slides are prepared to study under light microscope if slides were made please mention the procedure. Stereomicroscope or dissecting microscope are used for morphological studies if authors use this type of microscope then mention the model.

• I would like to see brief discussion of the area where the new species are found. Does it have interesting plant communities or unusual geology? Are there other narrowly endemic species of Caryophyllaceae in Vietnam and surrounding regions? Where are they found? A few sentences about these two topics would add to the interest of the article for a wider audience.

• Holotype of new species should be mentioned along with the illustration of new species.

• Taxonomic keys are also important in description of new species these are missing from this article.

Reviewer #2: The MS is well written and well conducted with methodology, results and discussions. The title is clear and informative covering morphological and molecular aspects and the background provides good context. However. i would suggest some points to be added in final draft

Molecular data lacs DNA extraction figures, markers amplification of PCR and sequencing.

Relative efficiency of rps16 and other markers for phylogeny.

Phyllograms based of molecular markers.

6. PLOS authors have the option to publish the peer review history of their article (what does this mean?). If published, this will include your full peer review and any attached files.

Reviewer #1: No

Reviewer #2: No

---

## [Author Response · Author response to Decision Letter 0]

7 May 2024

We have responded to all the information requested by Academic editors and reviewers in the attached file, which is Response to Reviwers. I hope that my revised manuscript can satisfy your requirements. If any please let me know. Thank you very much for your help.

Sincerely yours,

On behalf the authors

Tran Van Tien

---

## [Decision Letter · Decision Letter 1]

24 May 2024

Molecular Phylogeny and Cryptic Morphology: A Combined Approach to Taxonomic Novelties in Polycarpaea (Caryophyllaceae) from Vietnam

PONE-D-24-10601R1

Dear Dr. Tien Van Tran,

We’re pleased to inform you that your manuscript has been judged scientifically suitable for publication and will be formally accepted for publication once it meets all outstanding technical requirements.

Kind regards,

Waqas Khan Kayani, PhD

Academic Editor

PLOS ONE

Additional Editor Comments (optional):

Reviewers' comments:

Reviewer's Responses to Questions

**Comments to the Author**

1. If the authors have adequately addressed your comments raised in a previous round of review and you feel that this manuscript is now acceptable for publication, you may indicate that here to bypass the “Comments to the Author” section, enter your conflict of interest statement in the “Confidential to Editor” section, and submit your "Accept" recommendation.

Reviewer #1: All comments have been addressed

2. Is the manuscript technically sound, and do the data support the conclusions?

Reviewer #1: Yes

3. Has the statistical analysis been performed appropriately and rigorously? 

Reviewer #1: N/A

4. Have the authors made all data underlying the findings in their manuscript fully available?

Reviewer #1: Yes

5. Is the manuscript presented in an intelligible fashion and written in standard English?

Reviewer #1: Yes

6. Review Comments to the Author

Reviewer #1: (No Response)

7. PLOS authors have the option to publish the peer review history of their article (what does this mean?). If published, this will include your full peer review and any attached files.

Reviewer #1: No

---

## [Editor Report · Acceptance letter]

14 Jun 2024

PONE-D-24-10601R1 

PLOS ONE

Dear Dr. Tran, 

I'm pleased to inform you that your manuscript has been deemed suitable for publication in PLOS ONE. Congratulations! Your manuscript is now being handed over to our production team.

Kind regards, 

on behalf of

Dr. Waqas Khan Kayani 

Academic Editor

PLOS ONE